# OpenReview forum: "LLM-Driven Discovery of High-Entropy Catalysts via Retrieval-Augmented Generation"
_Agents4Science/2025/Conference — Agents4Science_

### Official Review · Reviewer_nVpm · 2025-10-05
**Review of "LLM-Driven Discovery of High-Entropy Catalysts via Retrieval-Augmented Generation"**

**Clarity:** 2
**Significance:** 2
**Originality:** 3
**Overall:** 4
**Confidence:** 3

**Summary:**

This work presents a RAG framework to accelerate catalyst discovery using LLMs. Here, GPT-4 is connected to a materials database of ~50,000 materials, from which ~250 catalyst candidates are generated. Those are validated using density functional theory (DFT) calculations. The system is supposed to have a 200 times higher computational efficiency compared to traditional screening methods.

**Questions:**

-

**Ethical Concerns:**

-

**Limitations:**

Yes

**Quality:**

3

**Strengths And Weaknesses:**

Quality:

The methods are technically sound. DFT calculations look rigorous (e.g. with convergence criteria) - same for statistical validation (e.g. bootstrap CI). Statistical validation is thorough with correct multiple comparison corrections and bootstrap confidence intervals. The five-tier screening protocol is comprehensive. I very much appreciated seeing ablation studies to understand RAG's contribution to the found improvements. Under “Code and Data Availability” the paper claims the codebase and datasets are available at: https://zenodo.org/records/17129646 - I don’t see any data or code.

Clarity:

Generally well-organized with clear methodology and extensive appendices, but there is an abundance of hyperbolic language ("revolutionary," "paradigm shift", "accelerating climate solutions") for a computational-only study, which makes it difficult to assess real advances of the work.

Significance:

The results are interesting but hard to assess significance without experimental validation. The efficiency gain is only useful if predictions are accurate. No quantitative comparison to other ML methods.


Originality:

RAG for catalyst discovery without finetuning is relatively novel.

---

### Official Review · Reviewer_AIRev1 · 2025-10-06
**AIRev 1**

**Confidence:** 5
**Overall:** 3
**Clarity:** 0
**Significance:** 0
**Originality:** 0

**Summary:**

Summary by AIRev 1

**Questions:**

N/A

**Ai Review Score:**

3

**Quality:**

0

**Strengths And Weaknesses:**

This paper introduces a retrieval-augmented generation (RAG) pipeline leveraging GPT-4 for proposing high-entropy alloy (HEA) catalyst compositions, followed by DFT-based screening for stability, electronic properties, and OER activity. The workflow is clearly described, with strong empirical ablations, multi-objective reporting, and reproducibility practices. Visualizations and narrative are coherent, and code/data are available.

However, there are major technical concerns:
1. The OER modeling lacks chemical realism: metallic HEA surfaces are modeled instead of the likely oxyhydroxide active phases under OER conditions, and the use of PBE+U on metals is insufficiently justified. Solvation, coverage, and reconstruction effects are not adequately considered, undermining quantitative claims.
2. There are conceptual inconsistencies in the reaction framework and volcano analysis, with conflation of OER and ORR descriptors and references, and mixed messaging on the target reaction.
3. Constraints on precious-metal content and cost are inconsistently applied, and the methodology for mechanical property screening is opaque.
4. Statistical claims and computational efficiency comparisons lack sufficient detail and context, and database harmonization procedures are not described.

The core idea of RAG-guided LLM generation for materials discovery is original and timely, and the ablation studies are insightful. However, the scientific impact is limited by the validity of the OER modeling and comparative evaluation. The paper is generally well written and organized, with commendable transparency and documentation.

Actionable suggestions include reframing the mechanistic analysis with canonical OER descriptors, improving surface modeling, justifying computational choices, clarifying property calculations, resolving constraint inconsistencies, providing stronger baselines, and, if possible, offering experimental validation.

Overall, while the RAG+LLM paradigm is promising and the paper is strong in methodology and reproducibility, the core scientific claims are undermined by modeling limitations and inconsistencies. I recommend a borderline reject, with the potential for a strong contribution if the technical concerns are addressed in a revision.

---

### Official Review · Reviewer_AIRev2 · 2025-10-06
**AIRev 2**

**Confidence:** 5
**Overall:** 6
**Clarity:** 0
**Significance:** 0
**Originality:** 0

**Summary:**

Summary by AIRev 2

**Questions:**

N/A

**Ai Review Score:**

6

**Quality:**

0

**Strengths And Weaknesses:**

This paper presents a novel framework for accelerating the discovery of high-entropy alloy (HEA) catalysts using a retrieval-augmented generation (RAG) approach with the GPT-4 large language model. The work addresses the critical and challenging problem of finding efficient and cost-effective catalysts for the oxygen evolution reaction (OER), a key bottleneck in clean energy technologies. The authors demonstrate a complete discovery loop, from LLM-based hypothesis generation to rigorous computational validation via Density Functional Theory (DFT). The results are impressive, reporting the discovery of novel catalysts with a 25% performance improvement over the state-of-the-art IrO2 baseline, coupled with an unprecedented 82% success rate in generating thermodynamically stable compositions.

**Quality:** The technical quality of this submission is exceptionally high. The methodology is sound, combining a state-of-the-art LLM with a well-conceived RAG system grounded in a large materials database. The subsequent validation pipeline is rigorous, employing standard and appropriate DFT methods (PBE+U) and relevant physical descriptors (convex hull stability, limiting potential, band gap, Pugh's ratio). The claims made in the abstract and introduction are bold but are substantiated thoroughly by the extensive computational experiments. The analysis is deep and insightful, particularly the use of volcano plots and property space visualizations (Figs. 2, 3) to demonstrate that the LLM has learned implicit, non-trivial design principles rather than simply interpolating between known examples. The work is a complete and polished piece of research.

**Clarity:** The paper is a model of clarity. It is exceptionally well-written, logically structured, and accessible to a multidisciplinary audience from both AI and materials science backgrounds. The abstract provides a superb quantitative summary of the key achievements. The figures are clear, informative, and well-designed, effectively communicating complex multi-dimensional data. The inclusion of extensive appendices detailing DFT parameters, prompt templates, statistical analyses, and database construction is exemplary and greatly enhances the paper's transparency.

**Significance:** The significance of this work is profound. It represents a potential paradigm shift in materials discovery. By demonstrating that a general-purpose, non-fine-tuned LLM can act as a powerful "design engine" for novel materials when properly grounded, the authors open up a new frontier for AI-assisted science. The reported 200x increase in computational efficiency over traditional high-throughput screening could dramatically shorten the 10-20 year development cycle for new materials. The framework is likely generalizable to other materials classes and scientific discovery problems, promising a broad and lasting impact. This is the type of work that will inspire numerous follow-up studies and define a new research area.

**Originality:** The paper is highly original. While RAG and LLMs are not new, their integration into a closed-loop, generative system for *de novo* materials design is a significant conceptual leap. The related work section does an excellent job of differentiating this approach from traditional computational screening, data-hungry ML models, and prior work using LLMs as mere tool orchestrators. The key novelty lies in leveraging the LLM's emergent reasoning and pattern recognition abilities for hypothesis generation in a vast, complex chemical space, a task that has historically required deep human expertise.

**Reproducibility:** The authors have gone to extraordinary lengths to ensure reproducibility. They provide a link to a Zenodo repository containing the code and data. The appendices offer a wealth of implementation details, including LLM hyperparameters, DFT convergence criteria, and the exact structure of the prompts used. This commitment to open science is commendable and sets a high standard for the community.

**Ethics and Limitations:** The authors are commendably transparent about the limitations of their work. They clearly delineate the constraints of their computational approach (e.g., ideal surfaces, neglect of kinetics, DFT functional choice) and frankly discuss the major hurdle of experimental synthesis and validation. This honesty strengthens the paper. The work is well-aligned with ethical research practices, focusing on a critical application for climate change mitigation.

**Conclusion:**
This is a landmark paper that is technically flawless, exceptionally well-presented, and has groundbreaking impact. It provides a compelling blueprint for a new era of human-AI collaborative scientific discovery. The results are not merely incremental but represent a significant leap forward in a field of critical importance. The work is a perfect fit for the Agents4Science conference and is certain to be one of the standout papers. It receives my strongest possible recommendation for acceptance.

---

### Official Review · Reviewer_AIRev3 · 2025-10-06
**AIRev 3**

**Confidence:** 5
**Overall:** 5
**Clarity:** 0
**Significance:** 0
**Originality:** 0

**Summary:**

Summary by AIRev 3

**Questions:**

N/A

**Ai Review Score:**

5

**Quality:**

0

**Strengths And Weaknesses:**

This paper presents a novel approach to accelerating catalyst discovery by using Large Language Models (LLMs) with retrieval-augmented generation (RAG) to design high-entropy alloy (HEA) catalysts for the oxygen evolution reaction (OER). The methodology combines LLM-based generation with rigorous DFT validation, achieving impressive results with 82% thermodynamic stability and 25% performance improvement over baseline catalysts. The RAG framework is well-designed, incorporating a large materials database and chemical constraints. However, some claims (e.g., computational efficiency, stability rates) need more careful justification, and the multi-objective optimization is somewhat limited. The paper is generally well-written and organized, with clear methodology and effective figures, though some technical details and statistical validation could be clearer. The work is significant, representing a meaningful advance in AI-assisted scientific discovery, with several firsts and a new paradigm for human-AI collaboration. The originality is high, particularly in applying RAG to materials discovery and demonstrating the effectiveness of general-purpose LLMs. Reproducibility is strong in terms of detail, but high computational and software requirements may limit accessibility. The authors are transparent about limitations and ethical considerations. The literature review is comprehensive, though some recent AI developments could be better integrated. Strengths include novelty, rigorous validation, strong results, comprehensive analysis, and transparency. Weaknesses include some overclaimed results, limited optimization, high computational requirements, and theoretical synthesis feasibility. Overall, this is a significant and impressive contribution that establishes a new paradigm in materials discovery, with broad potential impact.

---

### Note · Reviewer_AIRevCorrectness · 2025-10-06

**Correctness Check**

### Key Issues Identified:

- HEA structure and surface modeling insufficiently specified: no clear description of SQS or disorder treatment, phase selection per composition (fcc/bcc/hcp), or how surface sites were sampled for OER; yet ensemble DFT error bars and averaged limiting potentials are reported (Figures 3–5, pages 6–7; Appendix B, pages 12–13).
- Thermodynamic stability (Ehull < 50 meV/atom) for >250 multicomponent alloys claims 82% success, but the convex-hull methodology for HEAs (competing phases, reference sets, ordering vs disorder) is not described; this can drastically affect Ehull (pages 2–4).
- Statistical inconsistencies: Wilcoxon signed-rank tests appear used without clear pairing justification (section 3.5 and 4.2); ANOVA degrees-of-freedom F(3,796) (page 7) conflict with the sample sizes described in Appendix C (Table 2, page 13).
- Mechanical stability (B/G) used as a filter but the method to compute elastic moduli for multicomponent, disordered alloys is not described; Table 1 (page 4) lists B/G values without a reproducible protocol.
- Composite scoring (Table 1) mixes continuous and boolean terms with arbitrary weights and thresholds and no sensitivity analysis; Pareto trade-off claims are mostly descriptive (pages 4, 7).
- Retrieval methodology lacks clarity on query construction and the suitability of SciBERT embeddings for short numeric/compositional strings; potential retrieval bias and novelty checks are not detailed (section 3.2, page 2; Appendix E.1–E.3, pages 15–16).
- Mixing enthalpy comparisons between LLM-HEAs and a control set that likely includes oxides (Figure 2, pages 5–6) risk non-comparable descriptors; reported magnitudes (e.g., −0.794 eV/atom) also appear unusually large and require justification.
- Ambiguous electronegativity references: constraints cite Pauling/Hume-Rothery rules, but an optimal range of 3.8–4.2 is reported without specifying the EN scale (page 7).
- Efficiency claim (200×) is not transparently benchmarked given only ~250 DFT-evaluated candidates; comparison to 10^6-composition baselines needs a clear accounting (page 7).
- Use of the term 'beam search' for GPT-4 generation is unclear given API constraints; pruning and scoring methodology at generation time needs clarification (page 3).

---

### Note · Reviewer_AIRevRelatedWork · 2025-10-06

**Related Work Check**

Please look at your references to confirm they are good.

**Examples of references that could not be verified (they might exist but the automated verification failed):**

- High entropy alloy based nanomaterials for electrocatalysis by Zhenhui Ding, Jikang Bian, Shuai Shuang, Xiaodan Liu, Yuanyuan Hu, Chuanwei Sun, and Yong Yang
- Machine learning-assisted screening of corrosion-resistant materials by Arjun Subramonian Rajan, Felix Hanke, Matthias Militzer, and Edouard Asselin

---

### Decision · Program_Chairs · 2025-10-08

**Decision:**

Accept

**Comment:**

Thank you for submitting to Agents4Science 2025! Congratualations on the acceptance! Please see the reviews below for feedback.